# Spectral Diagnostic Model for Agricultural Robot System Based on Binary Wavelet Algorithm

**DOI:** 10.3390/s22051822

**Published:** 2022-02-25

**Authors:** Weibin Wu, Ting Tang, Ting Gao, Chongyang Han, Jie Li, Ying Zhang, Xiaoyi Wang, Jianwu Wang, Yuanjiao Feng

**Affiliations:** 1Guangdong Laboratory for Lingnan Modern Agriculture, Guangzhou 510642, China; wuweibin@scau.edu.cn (W.W.); 20203163056@stu.scau.edu.cn (T.T.); 20202009003@stu.scau.edu.cn (C.H.); lj@stu.scau.edu.cn (J.L.); zying@stu.scau.edu.cn (Y.Z.); 2College of Engineering, South China Agricultural University, Guangzhou 510642, China; 3College of Natural Resources and Environment, South China Agricultural University, Guangzhou 510642, China; gaoting@scau.edu.cn (T.G.); wxy15813369545@163.com (X.W.); wangjw@scau.edu.cn (J.W.); 4Key Laboratory of Agro-Environment in the Tropics, Ministry of Agriculture, South China Agricultural University, Guangzhou 510642, China; 5Guangdong Engineering Research Center for Modern Eco-Agriculture and Circular Agriculture, Guangzhou 510642, China

**Keywords:** agricultural robotics, diagnosis model, hyperspectral image, binary wavelet algorithm

## Abstract

The application of agricultural robots can liberate labor. The improvement of robot sensing systems is the premise of making it work. At present, more research is being conducted on weeding and harvesting systems of field robot, but less research is being conducted on crop disease and insect pest perception, nutritional element diagnosis and precision fertilizer spraying systems. In this study, the effects of the nitrogen application rate on the absorption and accumulation of nitrogen, phosphorus and potassium in sweet maize were determined. Firstly, linear, parabolic, exponential and logarithmic diagnostic models of nitrogen, phosphorus and potassium contents were constructed by spectral characteristic variables. Secondly, the partial least squares regression and neural network nonlinear diagnosis model of nitrogen, phosphorus and potassium contents were constructed by the high-frequency wavelet sensitivity coefficient of binary wavelet decomposition. The results show that the neural network nonlinear diagnosis model of nitrogen, phosphorus and potassium content based on the high-frequency wavelet sensitivity coefficient of binary wavelet decomposition is better. The *R*^2^, *MRE* and *NRMSE* of nn of nitrogen, phosphorus and potassium were 0.974, 1.65% and 0.0198; 0.969, 9.02% and 0.1041; and 0.821, 2.16% and 0.0301, respectively. The model can provide growth monitoring for sweet corn and a perception model for the nutrient element perception system of an agricultural robot, while making preliminary preparations for the realization of intelligent and accurate field fertilization.

## 1. Introduction

With the rapid development of robot technology, it has been increasingly applied in the agricultural field [1]. Due to the high intensity of field operations and complex road conditions, research on robot path planning, road recognition and perception, automatic and optimized navigation and robot arm control are all prerequisites for realizing robot field work [2,3,4,5,6,7,8,9,10]. The weeding and harvesting function systems of field robots have been the most studied, while there is little research on crop disease and insect pest perception, nutrient element content diagnosis and precision fertilizer spraying function systems [10,11,12]. The diagnosis of the nutrient elements content can provide a basis for field robots to perceive the crop growth status. Sweet corn is a widely cultivated food crop because it is rich in vitamins, amino acids and so on, making it more and more popular [13,14]. Nitrogen, phosphorus and potassium in sweet corn are important nutrients for its growth, which have great influence on its growth, yield and quality [15,16,17]. Because the traditional diagnosis of maize nutrient elements adopts chemical methods, a large number of corn samples need to be picked, and complicated chemical testing is required, which takes a long time and will cause damage to the plant [18]. Hyperspectral imaging technology is a simple, rapid and non-destructive method for the detection of crop nutrient elements. This method has become an important means to obtain field information in the field of digital agriculture and has been widely used in the detection of nutrient information of crops such as corn, wheat, tea and so on [19,20,21,22,23]. In recent years, through spectral technology, researchers have used stepwise regression, principal component analysis, support vector basis, random forest algorithm, continuous wavelet transform and other methods to estimate the crop chlorophyll content, nutrient element content, water content and other indicators quickly and in a nondestructive manner [24,25,26,27,28,29,30,31].

Wei et al. conducted inverse analysis on the soil organic matter content and improved its diagnostic efficiency by using hyperspectral indicators [32]. Wang et al. conducted quantitative inversion of salt ion content by using the spectral characteristics of salt ions, which were extracted from reciprocal logarithm of reflectance (Log(1/R)), providing an effective tool for the diagnosis of soil salinity [33]. Based on hyperspectral imaging technology, Wu et al. established a multiple linear regression inversion model of soil moisture content (SMC), which can quickly and efficiently predict soil moisture content [34]. Han et al. constructed a prediction model of soil AS content by extracting the spectral second-derivative characteristic variables, providing scientific basis and technical reference for soil pollution monitoring [35]. Yu et al. processed hyperspectral rice data by discrete wavelet decomposition, successive projection and principal component analysis. On this basis, the characteristic variables of the nitrogen content were extracted, and an inversion model of the nitrogen content in japonica rice was established. Among the results, the inversion model based on discrete wavelet decomposition is the best [36]. Fan et al. comprehensively compared and analyzed the performance of different types of spectral variables in estimating maize leaf nitrogen content (LNC) through partial least squares regression and a random forest algorithm. The results show that the PLS model with optimal multispectral variables has a better fitting effect and is a more effective model to evaluate maize LNC [37]. The first and second derivative processing of spectral reflectance and the construction of normalized spectral vegetation index can improve the correlation between characteristic variables and detection target content. However, due to different absorption or reflection conditions of different detection targets and the large variation in feature bands, most of the spectral information cannot be characterized. Therefore, more different feature extraction methods need to be introduced to improve the signal-to-noise ratio of spectral data and increase the stability and accuracy of the model.

Wavelet analysis is a new signal processing tool, which can reduce the dimension of spectral data, separate the high and low frequency information and facilitate the detection of singular points. Chen et al. processed hyperspectral reflectance data of soil samples by combining continuous media removal and wavelet packet decomposition, which improved the correlation between spectral reflectance and petroleum hydrocarbons in soil [38]. Gu decomposed hyperspectral datauses by wavelet transform algorithm. Then, the high-frequency information decomposed by wavelet technology is coupled with a random forest algorithm, which can effectively improve the prediction accuracy of soil organic matter content [39]. Li et al. preprocessed spectral data by mathematical transformation, a continuous wavelet transform algorithm and a correlation analysis algorithm. After the characteristic bands were extracted and selected, the estimation model of chlorophyll content in the stems and branches of Pitaya fruit was established. The *R*^2^ value based on continuous wavelet transform is 0.678 and the root mean square error *RMSE* = 0.037 [40]. Zhang et al. reduced the hyperspectral noise and improved the performance of the hyperspectral estimation model of soil organic matter content by using the wavelet energy characteristic method. The wavelet energy feature method could not only improve the estimation accuracy of the SNR (Signal–Noise Ratio) and soil organic matter content but could also realize the reduction in the dimensions of hyperspectral soil data and reduce the model’s complexity [41]. Wang et al. separated soil spectral data into five scales of high-frequency and low-frequency data by binary wavelet technology, then extracted the best band combination to build a diagnostic model of organic matter content, which has good stability [42]. Huang et al. decomposed canopy reflectance and its first derivative into wavelet coefficients by using the continuous wavelet transform method. The corresponding wavelet sensitivity coefficients were extracted, and the canopy LAI estimation model of late ripening wheat was established. Compared with models based on different types of hyperspectral vegetation indices, the accuracy of the late ripening wheat canopy LAI estimation model based on continuous wavelet coefficient was significantly improved [43]. Yang et al. decomposed spectral data by multi-scale wavelet. After extracting wavelet coefficients, partial least squares regression and support vector regression were used to construct the estimation model of tea polyphenol content. Compared with the model built by single feature variable, the multi-feature fusion method can improve the accuracy of estimating tea polyphenol content [44]. In conclusion, the pretreatment of hyperspectral reflectance data by wavelet analysis method can improve the correlation between feature bands and detection targets and improve the stability and accuracy of the model. However, the wavelet analysis method is rarely applied in the field of plant nutrient elements detection, and the research on sweet corn mostly focuses on nitrogen content, without comprehensive analysis of the main nutrient elements of corn nitrogen, phosphorus and potassium content.

In this paper, the comprehensive effects of different nitrogen application levels on the accumulation and absorption of nitrogen, phosphorus and potassium contents in maize were studied. The correlation between spectral characteristic variables and nitrogen, phosphorus and potassium contents of sweet corn was analyzed; then, the estimation model of nutrient elements was established by linear, parabolic, logarithmic and exponential functions. Then, the hyperspectral data of sweet corn are decomposed by binary wavelet. After analyzing the correlation between nitrogen, phosphorus and potassium contents and the frequency wavelet coefficients, the partial least squares regression and neural network nonlinear diagnosis model of nitrogen, phosphorus and potassium contents were established by extracting the wavelet sensitivity coefficients. Meanwhile, the stability, accuracy and precision of the model were evaluated by *R*^2^, *MRE* and *NRMSE*. AHP (Analytic Hierarchy Process) is used to assign the weight of the three evaluation factors, which is convenient to calculate the score of each model and compare the comprehensive performance of each model. The neural network model based on the binary wavelet high-frequency sensitivity coefficient has better comprehensive performance. It is of great significance to establish a diagnostic model for the rapid, accurate and nondestructive detection of nitrogen, phosphorus and potassium contents in maize leaves, which can not only make preliminary preparations for field operation robots to perceive the growth status of maize but can also provide a basis for the dynamic management of precise fertilization and topdressing.

## 2. Materials and Methods

### 2.1. Method of Obtaining Sweet Corn Samples

In the farm maize mono-cropping plot without fertilization, the soil in the tillage layer was dried and mixed with 2 cm sieve as the standby test soil. After mixing, the basic physical and chemical properties of the test soil were determined as urea during the test period, and nitrogen content was 46%. Nitrogen treatment during the whole growth period was as follows: no nitrogen application: 0 kg N·hm^−2^ (N_0_); Low-nitrogen: 100 kg N·hm^−2^ (N_1_); High-nitrogen: 300 kg N·hm^−2^ (N_2_). When maize grew to coniferous stage, nitrogen fertilizer treatment was 30% of the whole growth period, that is: no nitrogen: 0 kg N·hm^−2^ (N_0_); Low-nitrogen: 30 kg N·hm^−2^ (N_1_); High-nitrogen: 90 kg N·hm^−2^ (N_2_). After maize was grown for a week, two or three leaves were taken as samples to measure the following indices: nitrogen, phosphorus and potassium contents and corresponding hyperspectral data.

### 2.2. Determination of Nutrient Elements and Acquisition of Hyperspectral Data

In this experiment, the contents of total nitrogen, total phosphorus and total potassium in corn leaves were determined by distillation, vanadium molybdenum yellow colorimetric method and flame photometric method after H_2_SO_4_-H_2_O_2_ discooking [45]. Figure 1 shows the nutritional element diagnostic test system of an agricultural robot. The hardware includes a hyperspectral camera, a mobile platform, a supplementary light and a camera obscura. The software platform includes SpecView collection software and ENVI (Environment for Visualizing Imagesdata processing software. The spectral data curve of corn leaves determined by this system is shown in Figure 2. Four points at the same position of each leaf were selected to collect data, and their average value was used as the spectral reflectance of the sample. The hyperspectral data were corrected by Formula (1):(1)$$I_{0}=\mathrm{lg}\;\left[\frac{\left(I-I_{D}\right)}{\left(I_{W}-I_{D}\right)}\right]$$
where *I*_0_ is the corrected hyperspectral data, *I* is the original hyperspectral data, *I*_W_ is the white board average hyperspectral data and *I_D_* is the blackboard average hyperspectral data.

### 2.3. Extraction of Hyperspectral Characteristic Variables

The original hyperspectral data have a low SNR, and the band information is redundant. At the same time, there is a certain correlation between the reflectance data of various bands. The accuracy of diagnostic models based on raw data is low [46,47]. Extracting spectral characteristic variables to establish a diagnostic model of nutrient element content can reduce the computational cost. In this paper, hyperspectral location variables, hyperspectral area variables and vegetation index variables were adopted to analyze the correlation between nitrogen, phosphorus and potassium contents. The meanings and calculation methods of each hyperspectral characteristic variable are shown in Table 1 [48,49,50,51].

### 2.4. Binary Wavelet Analysis

As a signal processing tool emerging in recent years, wavelet analysis has the characteristic of being multi-scale, which can gradually observe the signal from coarse to fine, and has the function of describing the local features of the signal, which is beneficial to the detection of singular points. The binary wavelet is the semi-discretization result of continuous wavelet transform [52]. Let the scale parameter *a* = 2^*j*^, *j*∈*z*, and the translation parameter *b* still take the continuous value, as shown in Formula (2). In this case, the binary wavelet transform definition of *f*(*t*) is shown in Formula (3):(2)$$\psi_{2^{j},b}(t)=2^{-j/2}\times \psi \left[2^{-j}\left(t-b\right)\right]$$
(3)$$WT_{f}(2^{j},b)=2^{-j/2}\int_{-\infty }^{+\infty }f\left(t\right)\times \psi \left[2^{-j}\left(t-b\right)\right]dt$$

Binary wavelet can effectively separate low-frequency information from high-frequency information and retain all information of the original signal *f*(*t*). The high-frequency signal is the detail information in the original information and the low-frequency signal is the macro information in the original information, which provides a new idea for spectral signal processing and analysis [53]. Since the changes in nutrient element contents in sweet corn in hyperspectral reflectance were relatively weak, binary wavelet based on db_2_, db_3_, db_4_ and db_5_ wavelet bases was adopted to analyze the spectral data. Five decomposition layers were used to process and analyze spectral data and extract characteristic information in the spectrum.

### 2.5. Modeling Method and Accuracy Verification

The correlation between spectral characteristic variables and nitrogen, phosphorus and potassium contents was analyzed and calculated by using Excel, SPSS, Origin and Matlab software; then, the correlation coefficient was adopted for evaluation. The spectral characteristic variables with high correlation coefficients were selected as independent variables to construct four diagnostic models of high nitrogen, phosphorus and potassium contents, which were linear, parabolic, exponential and logarithmic. After binary wavelet decomposition, correlation analysis was conducted between high- and low-frequency wavelet coefficients and nitrogen, phosphorus and potassium contents of sweet corn, and corresponding wavelet sensitivity coefficients were extracted. Then, partial least squares regression (PLS) and an artificial neural network were used to construct the diagnostic models of nitrogen, phosphorus and potassium contents in sweet corn.

In order to objectively reflect the modeling accuracy, two—thirds of the samples were selected for modeling and one-third for verification. Moreover, the modeling determination coefficient *R*^2^, Mean Relative Error (*MRE*) and Normalized Root Mean Square Error (*NRMSE*) were used to comprehensively analyze the stability, accuracy and accuracy of the model. The formulae are as follows:(4)$$MRE=\frac{100\%\sum_{i=1}^{n}|y_{i}-y_{i}^{\prime }|/y_{i}}{n}$$
(5)$$RMSE=\sqrt{\sum_{i=1}^{n}{\left(y_{i}-y_{i}^{\prime }\right)}^{2}/n}$$
(6)$$NRMSE=\frac{RMSE}{\sum_{i=1}^{n}y_{i}/n}$$
where *y_i_* denotes the measured value of nutrient element content; *y_i_*′ represents the predicted value calculated by the inversion model; *I* is the number of sweet corn sample; and n is the number of verified sample 24.

The larger the determination coefficient *R*^2^ is, the better the fitting degree of the model is, while the smaller the *MRE* and *NRMSE* values are, the higher the inversion model accuracy is. In order to better reflect the synthesis of various models, AHP was adopted in this study, and the opinions of 7 experts in the field of nondestructive crop testing were consulted. The weight of evaluation factors *R*^2^, *MRE* and *NRMSE* were 46.48%, 29.58% and 23.94%, respectively. As *R*^2^ is larger and closer to 1, the stability is better. Finally, the reciprocal of *R*^2^, *MRE* and *NRMSE* multiplied by the corresponding weights were taken as the final score. The lower the score, the better the model performance. The comprehensive score is called *T*, as shown in Formula (7):(7)$$T=\frac{1}{R^{2}}\times 46.48\%+MRE\times 29.58\%+NRMSE\times 23.94\%$$

## 3. Results and Discussion

### 3.1. Changes of Nitrogen, Phosphorus and Potassium Contents under Different Nitrogen Application Treatments

As shown in Figure 3a, with the increase in the nitrogen application rate, the nitrogen content in maize leaves increased first and then decreased, and there were significant differences in the nitrogen content between the nitrogen application levels of N_0_, N_1_ and N_2_ (*p* < 0.05). Compared with the N_1_ and N_2_ treatments, the nitrogen content of leaves under the N_1_ and N_2_ treatments increased, and the nitrogen content under the N_1_ and N_2_ treatments was 1.28 times and 1.13 times higher than that under N_0_, respectively. The nitrogen content of leaves at the N_1_ level was reduced compared with that at the N_2_ level, and the nitrogen content at the N_1_ level was 1.12 times that at N_2_ level. The results showed that an appropriate increase in the nitrogen application rate could promote the absorption and accumulation of nitrogen in maize leaves, while a high nitrogen application rate inhibited the accumulation of nitrogen in maize leaves, which significantly decreased the nitrogen accumulation rate and reduced the utilization rate of nitrogen fertilizer. As shown in Figure 3b, with the increase in the nitrogen application rate, the phosphorus content of maize decreased, and there were significant differences in phosphorus content between the N_0_ and the N_1_ and N_2_ levels (*p* < 0.05). Compared with N_1_ and N_2_, the phosphorus content of maize under the N_0_ treatment decreased, and the phosphorus content under the N_0_ treatment was 1.47 times of that under N_1_ and 1.93 times of that under N_2_. The phosphorus content at the N_1_ level was 1.31 times higher than that at the N_2_ level, and the phosphorus content at the N_1_ level was lower than that at the N_2_ level. The results showed that with the increase in the nitrogen application rate, the accumulation of phosphorus in maize decreased rapidly at first and then at a decreasing rate. As shown in Figure 3c, with the increase in the nitrogen application rate, the potassium content of maize decreased, and there were significant differences in potassium content between nitrogen application levels N_0_ and N_1_ and N_2_, respectively (*p* < 0.05); the potassium content under nitrogen application treatments N_0_ and N_1_ were 1.40 times of that under N_2_ and 1.37 times of that under N_1_. The results showed that applying a small amount of nitrogen fertilizer had little effect on the uptake and accumulation of potassium in maize, but the application of high amounts of nitrogen inhibited the uptake of potassium and made the accumulation decrease rapidly.

### 3.2. Correlation Analysis and Diagnostic Model of Nitrogen, Phosphorus and Potassium Contents and Spectral Characteristic Variables in Sweet Maize

The calculated correlation coefficient values and significance test results of each spectral characteristic variable and nutrient element content are shown in Table 2.

According to the correlation coefficient calculation results in Table 2, nitrogen content was significantly correlated with *D_b_*, *λ_b_*, *D_y_*, *λ_y_*, *D_r_*, *λ_r_*, *R_g_*, *λ_g_*, *R_r_*, *λ_r_*, *SD_b_*, *SD_y_*, *SD_r_*, *SD_g_*, *VI*_1_, *VI*_2_, *VI*_4_, *VI*_5_ and *VI*_6_, among which the correlation coefficient with *R_r_*, *λ_r_* and *VI*_2_ was higher. The absolute values were all above 0.74. P content was significantly correlated with characteristic variables except for *VI*_3_ and *VI*_7_, and the correlation coefficients with *D_r_*, *R_g_* and *SD_b_* were higher than 0.9. Potassium content was significantly correlated with other characteristic variables except for *VI*_1_, *VI*_3_, *VI*_4_ and *VI*_7_, and the correlation coefficient with *SD_b_*, *SD_y_* and *VI*_6_ was higher than 0.56.

The 72-sample data after removing abnormal data were randomly divided into 2 groups, among which 48 samples were used for modeling and 24 samples were used for verifying the model accuracy. According to the correlation coefficient between the spectral index and nutrient element content, *R_r_*, *λ_r_* and *VI*_2_ were selected as independent variables of the diagnostic model to construct the diagnostic model of nitrogen content. *D_r_*, *R_g_* and *SD_b_* were selected as independent variables to construct a diagnostic model of phosphorus content. *SD_b_*, *SD_y_* and *VI*_6_ were selected as independent variables to construct a diagnostic model of potassium content. Four common regression models, a linear model, a parabola model, an exponential model and a logarithmic model, were used to construct the diagnostic model. The *MRE*, *R*^2^ and *NRMSE* were used to evaluate the comprehensive stability, accuracy and accuracy of the diagnostic models, respectively, and the T value was used to compare the comprehensive performance of each model.

Table 3 shows the accuracy and verification results of the model. The following can be seen: (1) Among the diagnostic models constructed by nitrogen content and hyperspectral characteristic variables, the parabolic model with *R_r_* as the independent variable had the lowest T value, with its modeling *R*^2^ of 0.672, *MSE* of 5.39% and *NRMSE* of 0.093. It can estimate nitrogen content effectively. (2) Among the diagnostic models constructed by phosphorus content and hyperspectral characteristic variables, the linear model fitted with *SD_b_* as the independent variable had the lowest T value, with its modeling *R*^2^ of 0.835, *MSE* of 11.97% and *NRMSE* of 0.120. It can estimate phosphorus content stably and accurately. (3) Among the diagnostic models constructed by potassium content and hyperspectral characteristic variables, the parabolic model fitted with *SD_b_* as the independent variable had the lowest T value, with its modeling *R*^2^ of 0.432, *MSE* of 10.22% and *NRMSE* of 0.112. As the modeling *R*^2^ was less than 0.5, potassium content could not be estimated stably.

### 3.3. Binary Wavelet Modeling

Spectral curves were reconstructed based on the low-frequency (high-frequency) data of different scales. The decomposition results of db_5_ were shown in Figure 4, where Figure 4a was the low-frequency wavelet coefficient curve and Figure 4b was the high-frequency wavelet coefficient curve. It can be seen from Figure 4a that the low-frequency wavelet coefficient curve preserves the morphological characteristics of the original spectrum. However, with the increase in the scale, the absorption characteristics of the spectral curve gradually weaken in the bands of 560−580 nm, 600−700 nm and 780–840 nm. The degree of separation of high-frequency information in original spectral data by binary wavelet is gradually deepened. Figure 4b shows that the spectral curve fluctuates strongly around 480, 610, 670 and 820 nm, indicating that the original spectral data are sensitive to the fluctuation of nitrogen, phosphorus and potassium contents in sweet corn near the corresponding band. It can be seen from the above that binary wavelet can effectively separate the high-frequency information in the original spectral data and highlight the absorption and reflection characteristics in the sweet corn spectrum.

Figure 5 shows the correlation curves between the A_5_ low-frequency wavelet coefficient and the D_5_ high-frequency wavelet coefficient of the db_5_ wavelet decomposition and nutrient element content, respectively. It can be seen from Figure 5 that the fluctuation range of the correlation curve between low-frequency wavelet coefficient and nitrogen, phosphorus and potassium content is small, and the correlation coefficient is smaller than the high-frequency wavelet coefficient. The correlation between the high-frequency wavelet coefficients and the contents of nitrogen, phosphorus and potassium in sweet corn was wide, and the correlation coefficients were basically significant. The contents of nitrogen, phosphorus and potassium in sweet corn leaves are low, and the response to the spectrum is weak, which is mainly reflected in details. Furthermore, the sensitivity of the high-frequency wavelet coefficient to the content of each element is better than that of the low-frequency wavelet coefficient.

The high-frequency wavelet coefficients were significantly correlated with the nitrogen, phosphorus and potassium contents of sweet corn leaves. The wavelet sensitivity coefficients of nitrogen, phosphorus and potassium were extracted from the high-frequency wavelet coefficients, then the diagnostic models of nitrogen, phosphorus and potassium contents in sweet corn were constructed by partial least squares regression and neural network. The stability, accuracy and precision of the models were evaluated comprehensively by modeling the *R*^2^, *MRE* and *NRMSE*, respectively. The *T* values of each model were calculated to comprehensively compare the performance of the diagnostic models.

The partial least squares regression model and the neural network nonlinear model select three wavelet sensitivity coefficients with high correlation coefficients as independent variables. The neural network framework adopts a feedforward neural network with each sample characteristic variable as input. The hidden layer is set to 1 and the number of neurons is set to 10. Finally, the corresponding nutrient element content is output. The nonlinear model of neural network is shown in Table 4.The diagnostic model of nitrogen content took wavelet coefficients 448 and 450 in the D_4_ high-frequency band of db_3_ and 367 in the D_5_ high-frequency band of db_2_ as independent variables. The correlation coefficients between the 3 variables and nitrogen content were 0.916, 0.958 and 0.918, respectively, denoted as *X*_db3-D4__−448_, *X*_db3-D4__−450_ and *X*_db2-D5__−367_, respectively. The diagnostic model of the phosphorus content takes wavelet coefficients at 527, 486 and 482 of the D_5_ high-frequency band of wavelet db_3_ as independent variables. The correlation coefficients with phosphorus content were 0.929, 0.930 and 0.952, respectively, denoted as *X*_db3-D5__−527_, *X*_db3-D5__−486_ and *X*_db3-D5__−482_, respectively. The diagnostic model of the potassium content takes wavelet coefficients at 455, 608 and 706 of the D_5_ high-frequency band of db_2_ as independent variables. The correlation coefficients with potassium content were 0.828, 0.916 and 0.792, respectively, denoted as *X*_db2-D5__−455_, *X*_db2-D5__−608_ and *X*_db2-D5__−706_, respectively.

### 3.4. Comparison between Spectral Characteristic Variable Modeling Results and Binary Wavelet Modeling Results

Table 5 shows that using the traditional spectrum characteristic of the variable construction of the nitrogen, phosphorus and potassium content of the diagnosis model, the correlation coefficient is relatively small; the model of the comprehensive stability, accuracy and precision is low; and the sensitivity based on the binary wavelet decomposition of the wavelet coefficient of nitrogen, phosphorus and potassium content of the partial least-squares regression diagnostic model and the neural network nonlinear model as correlation coefficient increased significantly. The comprehensive performance of the model was significantly improved.

Table 6 shows the comparison between the optimal diagnostic model with spectral characteristic variables as independent variables, the partial least squares diagnosis model and the neural network nonlinear diagnosis model.Compared with the parabolic model based on *R_r_* as independent variable, the *R*^2^, *MRE* and *NRMSE* of the partial least squares regression model based on the sensitivity coefficient of binary wavelet and high-frequency wavelet improved by 34.8%, 62.7%, 75.48% and 28.14%, respectively. The *R*^2^, *MRE* and *NRMSE* of the neural network nonlinear model are increased by 44.94%, 69.39% and 78.70%, respectively, and their comprehensive performance is improved by 28.14%. Compared with partial least squares regression model, the neural network model *R*^2^ improved by 7.51%, the *MRE* decreased by 17.91%, the *NRMSE* decreased by 13.16% and the comprehensive performance improved by 7.71%. Compared with the linear model based on *SD_b_* as independent variable, the *R*^2^, *MRE* and *NRMSE* of the partial least squares regression model based on the high-frequency wavelet sensitivity coefficient of the binary wavelet were improved by 10.06%, 41.19%, 30.42% and 11.95%, respectively. The *R*^2^, *MRE* and *NRMSE* of the neural network nonlinear model are increased by 16.05%, 24.64% and 13.25%, respectively, and the comprehensive performance is improved by 14.42%. Compared with the partial least square regression model, the neural network model increased *R*^2^ by 5.44%, *MRE* by 2.81% and *NRMSE* by 24.67%, but the comprehensive performance of the neural network model increased by 2.80%.

Compared with the parabolic model for potassium content diagnosis constructed with *SD_b_* as the independent variable, the *R*^2^, *MRE*, *NRMSE* and comprehensive performance of the partial least squares regression model constructed based on the high-frequency wavelet sensitivity coefficient of binary wavelet are improved by 86.80%, 61.64%, 59.46% and 47.18%, respectively. The *R*^2^, *MRE* and *NRMSE* of the neural network nonlinear model are increased by 90.05%, 78.86% and 73.13%, respectively, and the comprehensive performance of the neural network nonlinear model is improved by 48.84%. Compared with the partial least squares regression model, the neural network model improved *R*^2^ by 1.73%, *MRE* by 44.90% and *NRMSE* by 33.7%, but the comprehensive performance of neural network model improved by 3.13%. The results show that the high-frequency wavelet coefficients separated by binary wavelet can effectively suppress the interference of noise information, improve the signal-to-noise ratio of spectral data and improve the correlation between wavelet coefficients and nitrogen, phosphorus and potassium content, and improve the stability, accuracy and accuracy of the diagnostic model.

## 4. Conclusions

In this study, the specific conclusions are as follows:(1)With the increase in the nitrogen application rate, the nitrogen content in maize leaves increased first and then decreased, indicating that an appropriate increase in the nitrogen application rate could promote the absorption and accumulation of nitrogen in maize leaves, while a high nitrogen application rate could inhibit the accumulation of nitrogen in maize leaves, which significantly decreased the nitrogen accumulation rate and reduced the utilization rate of nitrogen. The decrease in the phosphorus content in maize indicated that with the increase in the nitrogen application rate, the accumulation of phosphorus in maize decreased rapidly at first and then at a decreasing rate. The decrease in the potassium content in maize indicated that the application of a small amount of nitrogen fertilizer had little effect on the absorption and accumulation of potassium in maize, and the application of a high amount of nitrogen would inhibit the absorption of potassium and make the accumulation decrease rapidly.(2)Binary wavelet can effectively improve the sensitivity of the spectrum to nitrogen, phosphorus and potassium contents of sweet corn and then improve the comprehensive performance of the model. Compared with the method of constructing spectral characteristic variables and vegetation incidices, it can effectively integrate the beneficial weak information in spectral data and suppress the influence of high-frequency noise. Compared with the parabola model based on Rr and the partial least squares regression model based on the binary wavelet high-frequency sensitivity coefficient, the comprehensive performance of the neural network nonlinear model based on the binary wavelet high-frequency sensitivity coefficient improved by 28.14% and 7.71%, respectively. Compared with the linear and partial least squares regression diagnosis models based on the high frequency sensitivity coefficient of binary wavelet, the comprehensive performance of the neural network nonlinear model based on the high frequency sensitivity coefficient of the binary wavelet improved by 14.42% and 2.80%, respectively. Compared with the parabola based on SDB as the independent variable and the partial least squares regression potassium content diagnosis model based on the binary wavelet high-frequency sensitivity coefficient, the comprehensive performance of the neural network nonlinear model based on the binary wavelet high frequency sensitivity coefficient is improved by 48.84% and 3.13%, respectively.(3)The chemical measurement method by using traditional destructive sampling of sweet corn nitrogen, phosphorus and potassium content and is sensitive to the high-frequency wavelet coefficient of building a neural network nonlinear sweet corn nitrogen, phosphorus and potassium content of the diagnosis model has good comprehensive performance, which can realize the rapid and nondestructive testing of sweet corn nitrogen, phosphorus and potassium content.

The diagnostic accuracy of nutrient element contents in this study can be further studied in order to improve the accuracy of agricultural robots to perceive nutrient element content in crops. The next step is to improve the SNR of the spectral data and improve the accuracy of the diagnostic model by using artificial neural network algorithm. Finally, the method can be applied to the nutrient element content sensing system of agricultural robot.

## Figures and Tables

**Figure 1 sensors-22-01822-f001:**
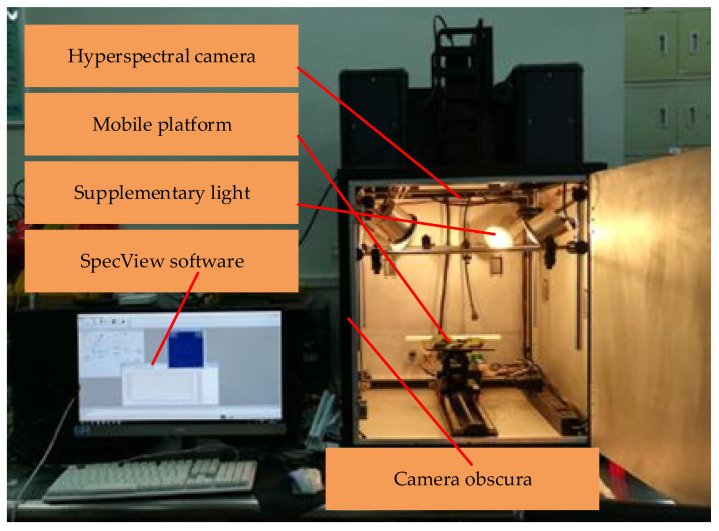
Agricultural robot system of hyperspectral imager.

**Figure 2 sensors-22-01822-f002:**
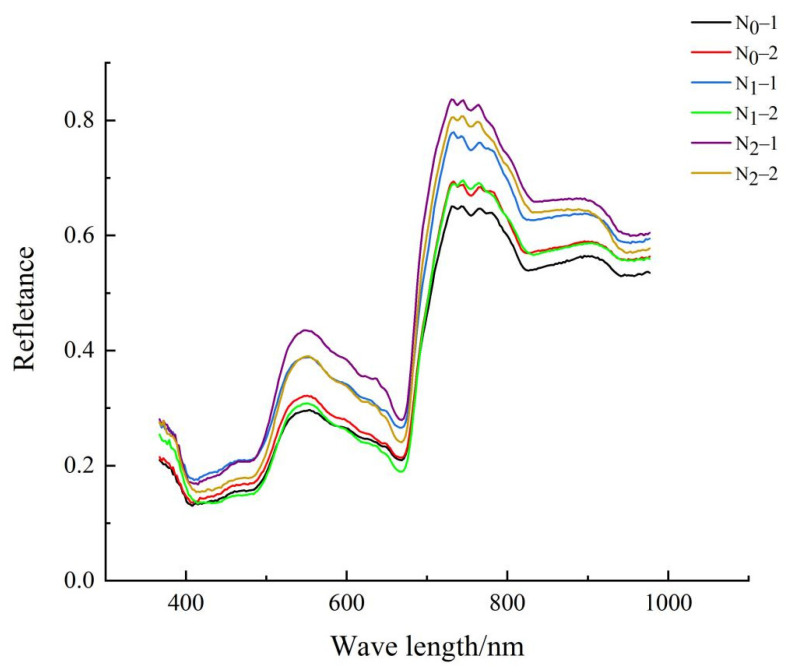
Spectral graphs of 6 samples.

**Figure 3 sensors-22-01822-f003:**
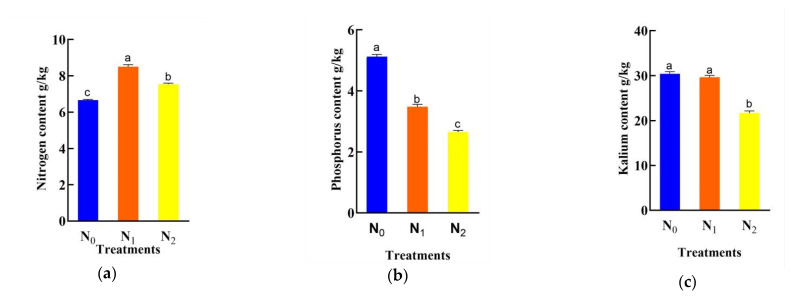
Significant analysis chart of nitrogen, phosphorus and potassium contents in maize under different nitrogen application treatments: (**a**) Significance results of nitrogen content in maize; (**b**) Significance results of phosphorus content in maize; (**c**) Significance results of potassium content in maize. Note: In the bar chart, “a, b, c” indicates that *p <* 0.05, they are arranged from large to small. Different letters indicate significant, and the same letters indicate insignificant.

**Figure 4 sensors-22-01822-f004:**
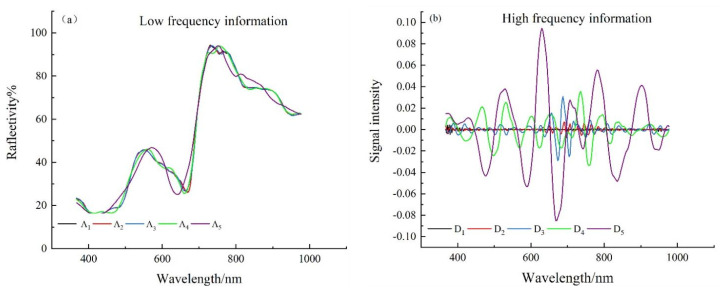
(**a**) Db_5_ wavelet analysis-low frequency information graph (**b**) Db_5_ wavelet analysis-high frequency information graph.

**Figure 5 sensors-22-01822-f005:**
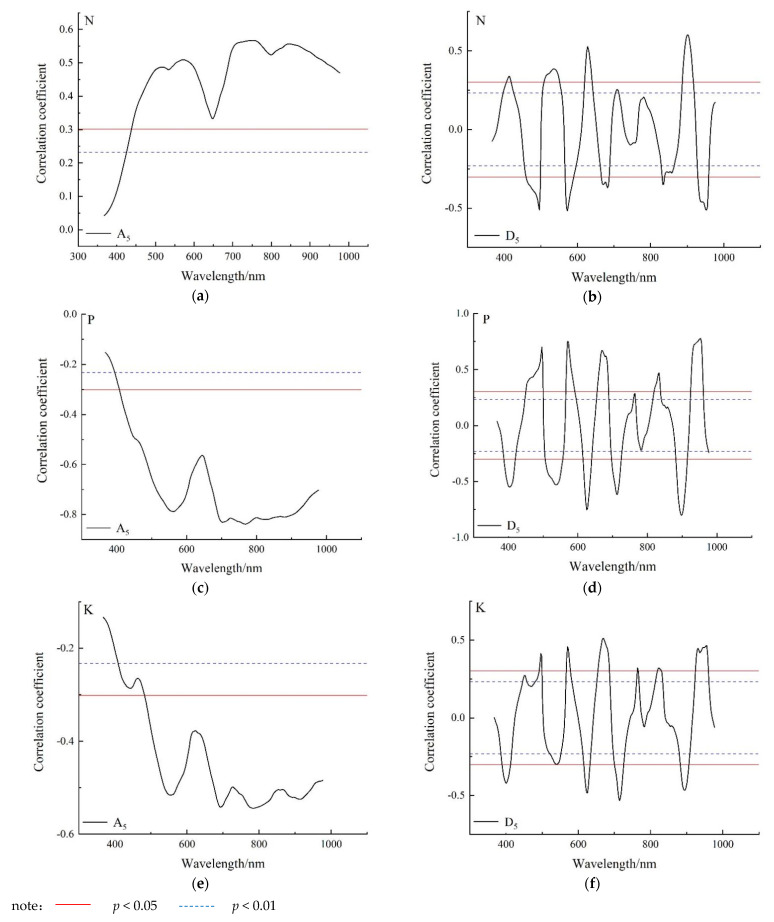
Correlation analysis of nitrogen, phosphorus and potassium contents with db_5_ low-frequency wavelet coefficients (A_5_) and high frequency wavelet coefficients (D_5_). (**a**) Correlation curve between nitrogen content and low frequency A_5_ wavelet coefficients. (**b**) Correlation curve between nitrogen content and D_5_ high frequency wavelet coefficient. (**c**) Correlation curve between phosphorus content and low frequency A_5_ wavelet coefficients. (**d**) Correlation curve between phosphorus content and high frequency D_5_ wavelet coefficients. (**e**) Correlation curve between potassium content and low frequency information A_5_. (**f**) Correlation curve between potassium content and high frequency information D_5_.

**Table 1 sensors-22-01822-t001:** Hyperspectral characteristic parameters and description.

Types of Spectral Characteristic Variables	Spectral Characteristic Variables	Parameter Description
Spectral position variable	Amplitude of blue edge *D_b_*	Maximum first-order differential spectral values at 490–530 nm
	Location of blue edge *λ_b_*/nm	The wavelength position corresponding to the blue amplitude
	Amplitude of yellow edge *D_y_*	Maximum first-order differential spectral values at 560–640 nm
	Location of yellow edge *λ_y_*/nm	The wavelength position corresponding to the yellow amplitude
	Amplitude of red edge *D_r_*	Maximum first-order differential spectral value within 680–760 nm
	Location of red edge *λ_r_*/nm	The wavelength position corresponding to the amplitude of the red side
	Green peak reflectance *R_g_*	Maximum first-order differential spectral value at 510–560 nm
	Green peak position *λ_g_*/nm	Wavelength position corresponding to the green peak reflectivity
	Red valley reflectance *R_r_*	Minimum first order differential spectral value within 650–690 nm
	Red valley location *λ_o_*/nm	Wavelength position corresponding to red Valley reflectivity
Spectral area variable	Blue edge area *SD_b_*	The area enclosed by the original light spectrum curve at 490–530 nm
	Yellow edge area *SD_y_*	560–640 nm spectrum curves surround the area of original light
	Red edge area *SD_r_*	The area enclosed by the original spectral curve within 680–760 nm
	Green peak area *SD_g_*	The area enclosed by the original light spectrum curve in 510–560 nm
Vegetation index variable	*VI*_1_ = *R_g_*/*R_r_*	Ratio of green peak reflectance to red valley reflectance
	*VI*_2_ = (*R_g_* − *R_r_*)/(*R_g_* + *R_r_*)	Normalized values of green peak reflectance and red valley reflectance
	*VI*_3_ = *SD_r_*/*SD_b_*	Ratio of the area of the red side to the area of the blue side
	*VI*_4_ = *SD_r_*/*SD_y_*	Ratio of the area of the red side to the area of the yellow side
	*VI*_5_ = (*SD_r_ − SD_b_*)/(*SD_r_ + SD_b_*)	The normalized value of the red-side area and the blue-side area
	*VI*_6_ = (*SD_r_ − SD_y_*)/(*SD_r_ + SD_y_*)	The normalized value of the area of the red and yellow sides
	*VI*_7_ = *R*_800_/*R*_680_	Simple ratio index SRI
	*VI*_8_ = (*R*_750_/*R*_720_) − 1	Red edge model REM
	*VI*_9_ = (*R*_750_ − *R*_445_)/(*R*_705_ − *R*_445_)	Correction of simple ratio index mSR_705_
	*VI*_10_ = (*R*_750_ − *R*_445_)/(*R*_750_ + *R*_705_ − 2 × *R*_445_)	Revised normalized difference index mND_705_

**Table 2 sensors-22-01822-t002:** Correlation coefficients of spectral characteristic variables with nitrogen, phosphorus and potassium contents.

Types of Variables	Nitrogen Content	Phosphorus Content	Potassium Content
*D_b_*	0.725 **	0.905 **	−0.543 **
*λ_b_*	0.732 **	−0.882 **	−0.507 **
*D_y_*	0.722 **	−0.908 **	−0.546 **
*λ_y_*	−0.713 **	0.863 **	0.485 **
*D_r_*	0.735 **	−0.908 **	−0.528 **
*λ_r_*	0.753 **	−0.826 **	−0.371 **
*R_g_*	0.722 **	−0.908 **	−0.546 **
*λ_g_*	0.735 **	−0.880 **	−0.495 **
*R_r_*	0.742 **	−0.899 **	−0.521 **
*λ_r_*	0.442 **	−0.569 **	−0.349 **
*SD_b_*	0.696 **	−0.913 **	−0.577 **
*SD_y_*	−0.654 **	0.896 **	0.579 **
*SD_r_*	0.734 **	−0.901 **	−0.519 **
*SD_g_*	0.697 **	−0.903 **	−0.558 **
*VI* _1_	0.343 **	−0.410 **	−0.171
*VI* _2_	−0.742 **	0.881 **	0.506 **
*VI* _3_	0.210	−0.228	−0.125
*VI* _4_	−0.406 **	0.545 **	0.291
*VI* _5_	−0.725 **	0.899 **	0.529 **
*VI* _6_	0.674 **	−0.899 **	−0.568 **
*VI* _7_	−0.009	0.016	0.014
*VI* _8_	−0.139	0.468 **	0.427 **
*VI* _9_	−0.219	0.504 **	0.409 **
*VI* _10_	−0.223	0.508 **	0.413 **

Note: ** means significant at 0.01 level.

**Table 3 sensors-22-01822-t003:** Results of nitrogen, phosphorus and potassium content diagnostic models.

Index to Be Predicted	Spectral Characteristic Variables	Model	Coefficient of Determination of Modeling *R*^2^ (*n* = 48)	Validation (*n* = 24)		
				*MRE*	*NRMSE*	*T*
Nitrogen content	*R_r_*	Linear	0.606	5.18%	0.090	0.8038
		Parabolic	0.672	5.39%	0.093	0.7298
		Index	0.639	5.18%	0.090	0.7643
		Logarithmic	0.631	5.08%	0.090	0.7731
	*λ_r_*	Linear	0.641	5.59%	0.091	0.7634
		Parabolic	0.667	9.32%	0.132	0.7559
		Index	0.667	6.05%	0.093	0.7369
		Logarithmic	0.644	5.58%	0.090	0.7599
	*VI* _2_	Linear	0.622	5.35%	0.092	0.7851
		Parabolic	0.665	5.23%	0.090	0.7359
		Index	0.650	5.20%	0.092	0.7524
		Logarithmic	0.657	4.71%	0.088	0.7424
Phosphorus content	*D_r_*	Linear	0.820	11.78%	0.119	0.6301
		Parabolic	0.821	11.83%	0.118	0.6294
		Index	0.755	11.55%	0.119	0.6784
		Logarithmic	0.812	18.46%	0.186	0.6715
	*R_g_*	Linear	0.820	11.70%	0.119	0.6299
		Parabolic	0.820	11.70%	0.119	0.6299
		Index	0.755	11.39%	0.118	0.6777
		Logarithmic	0.813	11.80%	0.126	0.6367
	*SD_b_*	Linear	0.835	11.97%	0.120	0.6208
		Parabolic	0.835	12.01%	0.121	0.6210
		Index	0.777	11.55%	0.118	0.6606
		Logarithmic	0.829	11.89%	0.120	0.6247
Potassium content	*SD_b_*	Linear	0.310	12.11%	0.132	1.5667
		Parabolic	0.432	10.22%	0.112	1.1329
		Index	0.307	39.00%	0.414	1.7285
		Logarithmic	0.282	12.32%	0.148	1.7200
	*SD_y_*	Linear	0.296	11.68%	0.129	1.6356
		Parabolic	0.324	11.05%	0.121	1.4962
		Index	0.293	11.39%	0.127	1.6504
		Logarithmic	0.315	11.31%	0.125	1.5389
	*VI* _6_	Linear	0.278	12.83%	0.138	1.7430
		Parabolic	0.279	12.88%	0.139	1.7372
		Index	0.272	12.47%	0.136	1.7784
		Logarithmic	0.279	12.90%	0.139	1.7373

**Table 4 sensors-22-01822-t004:** Results of the partial least squares diagnostic model for nitrogen, phosphorus and potassium contents of sweet corn based on binary wavelet sensitivity coefficient.

Index to Be Predicted	Partial Least Squares Regression Model	Coefficient of Determination of Modeling *R*^2^ (*n* = 48)	Validation (*n* = 24)		
			*MRE*	*NRMSE*	*T*
Nitrogen content	*Y* = 760.852 × *X*_db3-D4–448_ + 579.046 × *X*_db3-D4–450_ + 555.147 × *X*_db2-D5__−__367_ + 7.325	0.906	2.01%	0.0228	0.5244
Phosphorus content	*Y* = −55.083 × *X*_db3-D5__–__527_ + 39.259 × *X*_db3-D5–486_ + 20.589 × *X*_db3-D5–482_ + 9.124	0.919	7.04%	0.0835	0.5466
Potassium content	*Y* = 239.24 × *X*_db2-D5–455_ + 545.218 × *X*_db2-D5–608_ + 611.15 × *X*_db2-D5–706_ + 43.260	0.807	3.92%	0.0454	0.5712

**Table 5 sensors-22-01822-t005:** Results of neural network diagnosis model based on binary wavelet sensitivity coefficient for sweet corn nitrogen, phosphorus and potassium contents.

Index to Be Predicted	Coefficient of Determination of Modeling *R*^2^ (*n* = 48)	Validation (*n* = 24)		
		*MRE*	*NRMSE*	*T*
Nitrogen content	0.974	1.65%	0.0198	0.4868
Phosphorus content	0.969	9.02%	0.1041	0.5313
Potassium content	0.821	2.16%	0.0301	0.5412

**Table 6 sensors-22-01822-t006:** Comparison of spectral characteristic model, partial least squares model and neural network model results.

Index to Be Predicted	Model Type	Coefficient of Determination of Modeling *R*^2^ (*n* = 48)	*MRE*	*NRMSE*	*T*
Nitrogen content	*R_r_* parabolic model	0.672	5.39%	0.093	0.7298
	Partial least squares regression model	0.906	2.01%	0.0228	0.5244
	Neural network nonlinear model	0.974	1.65%	0.0198	0.4868
Phosphorus content	*SD_b_* Linear model	0.835	11.97%	0.120	0.6208
	Partial least squares regression model	0.919	7.04%	0.0835	0.5466
	Neural network nonlinear model	0.969	9.02%	0.1041	0.5313
Potassium content	*SD_b_* Parabolic model	0.432	10.22%	0.112	1.1330
	Partial least squares regression model	0.807	3.92%	0.0454	0.5984
	Neural network nonlinear model	0.821	2.16%	0.0301	0.5797

## Data Availability

Not applicable.

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
