# Peer review of "Spectral Diagnostic Model for Agricultural Robot System Based on Binary Wavelet Algorithm"

_sensors, 2022, doi:10.3390/s22051822_

Round 1
Reviewer 1 Report
In this manuscript, the authors study the effect of Nitrogen application rate on the absorption and accumulation of nitrogen, phosphorus and potassium contents on crops, using spectral characteristic variables and wavelet decomposition.
The paper is interesting, but there is a number of issues that the authors should clarify. It is important that the reader can fully understand and reproduce the methods and that the authors completely support the conclusions.
In some passages throughout the paper, the authors refer to wavelet analysis as a new signal processing tool. However, the wavelet analysis has been used in numerous research works and in a big number of applications. The authors should avoid this kind of sentences, or write them in such a way that they do not lead to confusion.
The quality of some figures is not good. In general, the authors should use a font size in the figures that make them readable (similar to the font size of the text of the manuscript). In figure 2, the colors cannot be easily distinguished. The same happens in fig. 4. In this figure, please clearly describe the meaning of the symbols in the legend.
In the first paragraph of section 2.1, the authors define 3 variables (N_0, N_1 and N_2), but each of these variables can mean two different things. This may lead to confusion, for example, in the contents of figure 2. Please clarify.
Please include horizontal lines in table 1 to clearly separate the variables that belong to each of the three concepts of the first column.
In general, in section 3, from figures such as fig. 3, I find it difficult to arrive to general conclusions, due to the low number of treatments (N_0, N_1, N_2) that the authors have considered in their study. For example, to support a sentence such as the one in lines 236-237, a higher number of treatments seems to be necessary.
In table 2, which threshold do the authors consider to say that nitrogen (or phosphorus or potassium) content is correlated with a specific variable?
In line 277, what do authors refer to with 'abnormal data'?
The number of samples in table 3 and next ones seems too low to arrive to general conclusions. Please justify.
I have missed concise information about the neural network architecture, training and configuration so that the experiments can be reproducible.
Please, clearly describe the meaning of all the acronyms the first time that they appear in the text.
The authors emphasize along the paper that this research work can be framed in the field of agricultural robotics. This is not a critical issue, but while it is true that the model that they propose could be integrated, in a future, in an agricultural robot, the current contents of the paper are more related with sensing and modeling in the agricultural field, and the relation with robotics is not as clear as the authors say.
The authors must carefully proofread the paper, as one can find numerous issues with the use of English language. There are also some typos. Among others:
- Please revise line 175.
- Please revise the sentence starting in line 198 (a verb is necessary).
- Please revise line 368.
Reviewer 2 Report
The reviewed paper aims to improve the perpetual (robot) sensing of nutrient element content based on a binary wavelet algorithm.
The scope of the paper and the conducted experiment are clear, scientifically sound and accompanied by a comprehensive description. Connection to the scope of the Sensors journal seems rather loose. Nevertheless, I consider it still appropriate, with respect to the special issue into which this manuscript has been submitted.
As such, I provide the authors only the following comments:
- The introduction provides a sufficient state-of-the-art with only one exception. The contemporary challenging tasks of optimal navigation in (precision) agriculture is omitted. I recommend, e.g. the following references for the introduction:
- Reznik, T.; Herman, L.; Klocova, M.; Leitner, F.; Pavelka, T.; Leitgeb, S.; Trojanova, K.; Stampach, R.; Moshou, D.; Mouazen, A.; et al. Towards the Development and Verification of a 3D-Based Advanced Optimized Farm Machinery Trajectory Algorithm. Sensors 2021, 21, 2980.
- Hameed, I.A. Intelligent Coverage Path Planning for Agricultural Robots and Autonomous Machines on Three-Dimensional Terrain. J Intell Robot Syst 2014, 74, 965–983, doi:10.1007/s10846-013-9834-6.
- The paper's goals are somehow present; however, I would appreciate it if they would be explicitly listed at the end of section 1 (Lines 122-139). Moreover, I would prefer null hypotheses for the experiment, especially the conclusions (Lines 416-438), which seem close to a hypothesis-driven scientific paper.
- Has ever been similar research conducted? If so, then such a discussion is missing between lines 379-408. If not, then an explicit statement should be made at line 408.
- Lines 410-415 seem too much as an abstract rather than conlusions. I propose deleting them.
- There are many typos across the whole manuscript. These shed a bad light on the manuscript with clear added value.
Round 2
Reviewer 1 Report
The paper has improved with the revision.
However, there are still two concerns that remain:
- I keep on thinking that the number of treatments (N_0, N_1, N_2) that the authors consider in the study is too low to arrive to general conclusions. The authors have not increased the number of treatments during the revision, and they have not given any successful reason for not doing so.
- The authors describe the main features of the neural network architecture in the letter of response. However they say that since it is a conventional neural network, the description is not included in the paper. I do not agree with it. The paper should contain enough information so that the experiments could be replicated by any reader. The description of the architecture should be included in the paper and, additionally, as I commented in my previous report, the authors should also describe the training process.
